# Research Topic Trends on Turnover Intention among Korean Registered Nurses: An Analysis Using Topic Modeling

**DOI:** 10.3390/healthcare11081139

**Published:** 2023-04-15

**Authors:** Jung Lim Lee, Youngji Kim

**Affiliations:** 1Department of Nursing, Daejeon University, Daejeon-si 34520, Republic of Korea; leejl@dju.kr; 2Department of Nursing, College of Nursing and Health, Kongju National University, Kongju-si 32588, Republic of Korea

**Keywords:** nurses, employee turnover, data mining, social network analysis

## Abstract

This study aimed to explore research topic trends on turnover intention among Korean hospital nurses by analyzing the keywords and topics of related articles. Methods: This text-mining study collected, processed, and analyzed text data from 390 nursing articles published between 1 January 2010 and 30 June 2021 that were collected via search engines. The collected unstructured text data were preprocessed, and the NetMiner program was used to perform keyword analysis and topic modeling. Results: The word with the highest degree centrality was “job satisfaction”, the word with the highest betweenness centrality was “job satisfaction”, and the word with the highest closeness centrality and frequency was “job stress”. The top 10 keywords in both the frequency analysis and the 3 centrality analyses included “job stress”, “burnout”, “organizational commitment”, “emotional labor”, “job”, and “job embeddedness”. The 676 preprocessed key words were categorized into five topics: “job”, “burnout”, “workplace bullying”, “job stress”, and “emotional labor”. Since many individual-level factors have already been thoroughly investigated, future research should concentrate on enabling successful organizational interventions that extend beyond the microsystem.

## 1. Introduction

Because of population aging, the rise in patients with chronic diseases, and modifications to healthcare delivery systems, there is an increased demand for nurses. The coronavirus disease 2019 (COVID-19) pandemic has also drawn particular attention to nurse staffing. The shortage of nurses is a global issue, and the World Health Organization (WHO) estimates that the globe would require 9 million more nurses and midwives by 2030 [1]. Patient health outcomes including mortality, nosocomial infection rates, and safety incidents are impacted by the nursing shortage, as are nurse outcomes such as health and job satisfaction [2,3]. As a result, training and deploying competent nurses is a national obligation in order to improve public health and provide high-quality nursing care.

To address the global shortage of nurses, the WHO recommended increasing the number of nursing graduates by 8% each year until 2030 [1]. Republic of Korea responded to this issue by establishing more nursing departments throughout the country in 2008 and increasing admission quotas. The number of nurses trained at four-year colleges has since expanded dramatically, from 3724 in 2009 to 9434 in 2016 and is still growing [4]. However, given that only 46.7% of licensed nurses are now employed, which is lower than the Organization for Economic Co-operation and Development (OECD) average of 67.2%, there is a higher need for steps to keep nurses, such as reducing nurse turnover, than for initiatives to increase the supply [5].

Nurse turnover rates vary among countries, ranging from 15.1% in Australia [6] to 27.65% in the United States [7]. In 2020, the turnover rate of nurses in Korean hospitals was 14.5%, ranging from 13.6 to 29% depending on the size of the hospital [8]. The reasons for turnover included task maladaptation (17.7%), transfer to another hospital (12.2%), change to another job (12.2%), and health reasons (10.6%). In particular, the turnover rate of new nurses in Republic of Korea has increased dramatically from 27.8% in 2011 to 42.7% in 2018, and reaching 47.7% in 2020, indicating that nurse turnover has become a very serious problem [9]. Hence, in order to create a nurse staffing policy that works, it is necessary to analyze the variables that are linked to turnover or turnover intention.

Lee and Kang [10] conducted a systematic review of 263 articles published from 2006 to 2016 on the factors associated with turnover intention among Korean hospital nurses. They classified the factors using an ecosystemic perspective at the levels of the individual, family, occupation, ward, hospital, and system (society). Woodward and Willgerodt [11] also conducted a systematic review of 34 studies published from 2010 to 2020, on nurse turnover and employment in the United States, classifying the related factors as individual-level, unit-level, and organizational-level. Both studies showed that previous research had mostly looked at issues at the individual and ward levels, with little focus on hospital and system levels. While American nurses demonstrated associations with job embeddedness at the individual level and magnet designation at the organizational level, Korean nurses showed associations with emotional labor and role conflict at the family and occupational levels, as well as integrated nursing care services and hospital accreditation at the social level. Therefore, when examining factors linked to turnover intention, the socio-cultural context should be taken into account because the socioeconomic structure and healthcare systems differ between nations.

To effectively manage the nursing workforce, it is essential to develop human, physical, and institutional strategies in a multifaceted approach that allow nurses to stay in the field for longer periods of time, rather than simply increasing nurse staffing. In addition, multi-level approaches at individual, unit, organization, and nation levels should be considered. Studies on nurse turnover intention conducted so far have been focused on identifying the phenomenon, such as investigating the influencing factors and classifying the factors through systematic research [10,11]. These research methods have some limitations in analyzing and gaining insights on this topic in an integrated manner. Therefore, it is necessary to identify the research topic trends and present the direction of future research by analyzing and clustering the keywords shown in the latest studies.

Text network analysis is a method of analyzing the structure of words, their communication mechanisms, and their relationships in text data collected for research purposes, and building a network is useful for visualizing texts’ overall structure and understanding them holistically [12]. Topic modeling is a representative method of text network analysis used in various academic fields, including nursing, that is suitable for analyzing research topic trends by locating keywords and topics in each document, and then clustering and analyzing possible topics [13]. Therefore, this study aimed to identify the main topics covered in papers associated with turnover or turnover intention of Korean nurses using topic modeling, and objectively and clearly examining the research topics inherent in the text of the literature.

## 2. Materials and Methods

### 2.1. Study Design

This descriptive study employed text network analysis and topic modeling to extract keywords and topics from titles, abstracts, and author keywords of studies related to nurse turnover in Republic of Korea, facilitating the identification of research topic trends.

### 2.2. Subjects

The study subjects included the titles, abstracts, and author keywords of research papers on the turnover and turnover intention of Korean nurses published in domestic and international journals. Letters to the editor, conference presentation papers, and dissertations that did not go through editorial or peer review processes were excluded.

### 2.3. Data Collection

Based on the previous study, research papers published from January 2010 to June 2021 were collected to identify research topic trends over the past decade [11]. Research papers published from January 2010 to June 2021 were collected to identify research topic trends over the past decade. The search terms “nurse” and “turnover” were used, and 335 articles from Nurimedia (DBPIA), 322 from National Science and Technology Information Center, 464 from Research Information Sharing Service, and 110 from the Korean Studies Information Service System were collected from four Korean databases. Additionally, 104 articles from PubMed and 83 articles from CINAHL were collected through international databases. The collected articles were transferred to RefWorks, a bibliographic information program, to eliminate duplicate papers. A total of 976 papers—794 domestic papers and 182 overseas papers—were then reviewed by hand. Additionally, 586 papers were excluded since the subjects were not nurses, the content was not associated with turnover, or the papers were considered duplicates, resulting in the selection of 390 papers.

### 2.4. Data Analyses

A flow chart depicting the research process is presented in Figure 1. The number of papers published per year was computed, and the abstracts, author keywords, and titles of the papers were extracted. An automatic filter function was applied to extract only nouns. The NetMiner analysis program’s dictionary function (NetMiner v.4.4.3, Cyram Co. Ltd., Seongnam, Republic of Korea) was used to delete excluded words and specify synonyms to organize words. Compound nouns were designated in advance to be recognized as single words (e.g., job stress or job satisfaction) through a literature review. After word refinement, a network was created by extracting words with a TF–IDF (the term frequency–inverse document frequency) of 0.5 or higher and 2 characters or more to use words that appear more often in a specific document than in multiple documents. The TF is the number of times a particular word appears in a particular document. The IDF is the reciprocal of the frequency of a particular word in all documents. In other words, the TF–IDF index is the weighted frequency obtained by applying a weight to a simple frequency, and it shows how much weight a particular word has in a document. The TF–IDF index is used to evaluate the topic of an article based on the words it contains, since its use of weights enables it to measure relevance, not frequency [14]. The frequency of the simultaneous occurrence of words was calculated, and semantic network analysis was performed accordingly.

Latent Dirichlet allocation (LDA)-based topic modeling was performed to extract topics and identify keywords with a high allocation probability by topic. LDA-based topic modeling analyzes unstructured text to find meaningful and important topics, allowing context-related topics to be inferred from vast unstructured texts [15]. This analysis method has the advantage of quantitatively finding meaning from large amounts of data, and it is actively used in various fields, including nursing, library and information science, and business administration. To obtain meaningful results using topic modeling, the number of topics should be determined by the researcher. In this study, several simulations were performed to generate an optimal categorization without overlap. The basic settings for determining the suitable number were an alpha value of 0.01, a beta value of 0.01, and a sampling repetition count set to 1000. The experiment was repeated with the number of topics set to range from 4 to 10 [16]. After a comparison of the results, the researchers in this study concluded that limiting the number of topics to five would provide the most meaningful and appropriate interpretation. When there were five topics, the topics were the most independent from one another and were considered to reflect the research topics most comprehensively. To visualize the network of words by topic, the top 50 words with high allocation probabilities for each topic were extracted to form a word topic network, and the keywords with high influence in those topics were depicted.

Centrality analysis was conducted through one-mode network analysis to identify the influence of keywords. Degree centrality, betweenness centrality, closeness centrality, and the frequencies of occurrence were analyzed. Degree centrality shows how many nodes in a network are connected, betweenness centrality reflects the degree to which a word plays a mediating role in a network, and closeness centrality shows how close a word is to other words [17].

## 3. Results

### 3.1. Publication Trends

Eight articles were published in 2010 and 16 in 2011, which was an increase of 100%. Since then, more than 20 articles have been published each year, with 50 articles published in 2020. However, there was a temporary drop in the number of articles published in 2021 through June (Figure 2).

### 3.2. Topic Modeling

In this study, topic modeling was performed using 676 words from the analysis. The number of topics with the most appropriate mutual independence of research topics was determined to be five. The researchers named each topic considering the associations among major keywords with a high probability of assignment for each research topic (Table 1). The labeled topics were “job”, “emotional labor”, “job stress”, “workplace bullying”, and “burnout”. Work-related keywords such as job, job embeddedness, work environment, and career were mostly derived in the “job” topic, while management and conflict were also derived in the “emotional labor” topic. The “job stress” topic addressed, in addition to job stress, support and fatigue. Workplace bullying and violence were mostly drawn from the “workplace bullying” topic, while organizational commitment and professionalism were also derived from the “burnout” topic. The visualized word–topic network, which was composed of words with a high influence by topic, shows that the word “role” mediated “job stress” and “emotional labor” (Figure 3).

### 3.3. Centrality Analyses

The word with the highest degree centrality was “job satisfaction”, the word with the highest betweenness centrality was “job satisfaction”, and the word with the highest closeness centrality and frequency was “job stress”. The following words were among the top 10 in the frequency analysis and all three centrality analyses: “job stress”, “burnout”, “organizational commitment”, “emotional labor”, “job”, and “job embeddedness” (Table 2).

As a result of classifying the keywords included in Table 2 according to the Ecological Systems Theory [18], most of the keywords were included in the microsystem including at the individual level. The number of included keywords decreased as the system was expanded to mesosystem, exosystem, and macrosystem (Figure 4).

## 4. Discussion

This study utilized topic modeling to examine the words in the abstracts, author keywords, and titles of 390 articles associated with turnover among Korean nurses published in the last 10 years. The analysis yielded important keywords associated with turnover and provided insights into major research topic trends. Based on the findings, we propose directions for future research that can assist in the establishment of strategies to reduce nurse turnover.

Through topic modeling, we identified five major topics, which were then ranked in descending order based on their likelihood of occurrence: “job”, “burnout”, “workplace bullying”, “job stress”, and “emotional labor”. According to the topics “burnout”, “workplace bullying”, “job stress”, and “emotional labor”, previous research has mostly concentrated on exploring factors that increase the likelihood of turnover or turnover intention. The topic “burnout” included keywords such as “organizational commitment”, “professionalism”, “shift”, and “role conflict”. “Job stress” included “stress”, “support”, “fatigue”, and “perception”. “Emotional labor” included “management”, “conflict”, “environment”, and “emotional intelligence”. The word that mediated between the topics “job stress” and “emotional labor” was found to be “role”. “Emotional labor” was not discovered in studies aimed at nurses in other countries [11,19], but it was derived in studies focusing on the turnover intention of Korean nurses, which may be attributed to sociocultural differences [10,20]. The inclusion of “burnout” as a reason for nurse turnover in the US over the past decade yielded similar results, although being distinct from this study’s major topics, “emotional labor” and “workplace bullying [11].

This study also differed from a meta-analysis of nurses in 12 countries, in which job-related factors such as “team cohesion”, “recognition”, and “work–family conflict” were found to be predictors of turnover intention [19]. This suggests that studies on the turnover intention of Korean nurses focused on personal stress or emotional labor at the individual level, which should be viewed in light of the cultural and societal environment. One explanation for this trend could be that turnover among Korean nurses is caused by the problems posed by the environment in which they perform their jobs, rather than by the profession itself. Therefore, it is critical to aggressively implement improvement initiatives at the organizational level to reduce emotional labor in the work environment of nurses.

“Role conflict” was a noteworthy keyword that was pertinent to the three areas of “burnout”, “work stress”, and “emotional labor”, and it had a substantial effect on the intention of Korean nurses to leave their jobs [10]. Only 6 out of 263 studies listed “role conflict” as a factor, according to Lee and Kang [10], but when topic modeling was applied to examine research patterns, it became a very significant keyword. “Role conflict” is widespread in positions of expertise that demand specific legal activity, necessitating the enactment of the Nursing Law. “Organizational commitment”, “professionalism”, “support”, and “management” are keywords connected to these three subjects that can reduce turnover intentions and be attained through efficient managerial leadership. Although “leadership” was not a keyword for this study, other research has shown that leadership is an important element in lowering turnover intention [10,11,19,21]. Leadership is essential not only at the ward and organizational levels but also in national positions as it can influence nursing policy decisions.

The topic of “workplace bullying” included keywords such as “violence”, “workplace”, “patient”, and “empowerment”. “Violence” has been identified as a factor that promotes turnover intention [10,11]. “Workplace bullying” is a comparatively recent concept that debuted in 2010 with a single article in Republic of Korea [13]. It emerged as an important topic associated with turnover intention in this study, which was based on research conducted over the past ten years. Workplace bullying is reported to occur at a rate of 21% in Republic of Korea [22] and 33.4% in Saudi Arabia [23], and it is particularly prevalent among new nurses and is closely related to the intention to leave the profession. In Republic of Korea, where there is a high rate of new nurse resignation, anti-bullying regulations should be adopted at the organizational level, and unit managers should show initiative to guarantee that these policies are effectively implemented in the field.

The topic of “job” included keywords such as “job embeddedness”, “work environment”, “career”, and “health”, providing strategic directions to assist nurses in staying on the job. Other topics, such as “emotional labor”, “job stress”, “workplace bullying”, and “burnout”, focus on finding the reasons and associated factors of turnover intention and belong to negative concepts. These topics demonstrate the substantial advancement in the field that academics have been researching for the past ten years. In the study of nurse turnover in the US from 2010 to 2020 [11], “job embeddedness” was identified as a keyword that contributed to factors at the individual level, but it was not discovered in studies on nurse turnover in Republic of Korea from 2006 to 2016 [10]. Current studies have focused on “job embeddedness”, which is considered a key concept in organizational retention. Further research is needed to fully comprehend the influence of job embeddedness in the community, also known as off-the-job embeddedness [24].

Using comparable keywords to those found in the “job” topic of this study, Nurdiana and Hariyati [25] proposed five nurse retention strategies, including (1) a favorable working environment; (2) leadership; (3) nurse staffing; (4) orientation, preceptorship, and mentorship; and (5) professional development. Although many previous studies conducted in the last ten years have focused on identifying the factors associated with the reasons why Korean nurses leave the workplace, future research should take a more interventional approach, focusing on positive topics such as the creation of a healthy work environment, career development, wellness, or nurses’ health.

“Job stress”, “burnout”, “organizational commitment”, “job embeddedness”, “stress”, “workplace bullying”, “conflict”, “professionalism”, “organizational culture”, “role conflict”, and “fatigue” were among the top 30 keywords in terms of frequency and all three types of centrality. “Job satisfaction”, “work”, “experience”, “satisfaction”, and “work–family” were among the top 30 keywords for all three categories of centrality but did not rank among the top 30 in terms of frequency.

Following that, we investigated “job satisfaction”, “organizational culture”, and “work–family” without relying on the topic modeling keywords. “Job satisfaction” is strongly related to turnover intentions and patient care quality, but because the factors influencing it can vary widely, it is necessary to identify the most influential factors [26]. Furthermore, the previous study indicates that improving internal factors at the individual level is the main goal of interventions to increase job satisfaction. Future interventions should, however, also concentrate on organizational improvements such enhancing the working environment [27]. “Organizational culture” is an important factor at the unit and organizational levels that, when enhanced, can control the factors causing turnover at the individual level [10]. Additionally, leadership is a critical factor in creating a healthy work environment [28]. “Work–family” is a factor at both the family and occupational levels, accounting for only 7.5% of the causes of turnover among Korean hospital nurses in 2021 [8,10]. Nevertheless, since 89% of nurses worldwide are women [1], and the average service years for nurses in Republic of Korea are 7 years and 8 months [9], it is essential to establish organizational and institutional support that can effectively create a “work–family” balance to prevent nurse turnover.

We classified the turnover-associated keywords according to the Ecological Systems Theory, and most of the keywords were included in the microsystem including at the individual level. However, strategic initiatives, such as Nursing Law, and national nursing leadership positions in the exosystem and macrosystem should be prioritized to address them, and the related factors should be explored accordingly. Furthermore, since most studies to date have focused on identifying factors, future studies should focus on uncovering causes and leading to interventions. To create a new nursing paradigm, we need to make efforts such as restructuring the job system for nurses’ career development. We should also undertake various intervention studies, such as expanding internships, orientation, and transition programs, to reduce turnover among new nurses, who account for the majority of turnover.

In this study, abstracts, author keywords, and titles of publications acquired through search engines were analyzed among research papers published in the last decade, with the caveat that the published studies may be biased. When topic modeling is conducted, the possibility cannot be ruled out that subjective criteria based on the researcher’s assessment are reflected in the refinement process of the keywords and post-analysis interpretation process, leading to an arbitrary interpretation [29]. Therefore, to enhance the validity of topic labeling in future research, including the analysis process, it is recommended to include a group of external experts in the topic naming process.

## 5. Conclusions

This study examined 300 papers on the turnover intention of Korean nurses published in the recent decade and discovered that the studies were mostly conducted at the individual and ward levels. Unstructured text keywords taken from abstracts, author keywords, and paper titles were refined and grouped using topic modeling into “job”, “burnout”, “workplace bullying”, “job stress”, and “emotional labor” in this study. The fact that topics such as “emotional labor” or “workplace bullying” were derived implies that the social environment has a large influence on the turnover of Korean nurses. Based on the Ecological Systems Theory, we discovered that research interest was concentrated on microsystems. Since many individual-level factors have already been thoroughly investigated, future research should focus on eliminating bullying and improving social awareness to facilitate more successful interventions on the derived issues.

## Figures and Tables

**Figure 1 healthcare-11-01139-f001:**
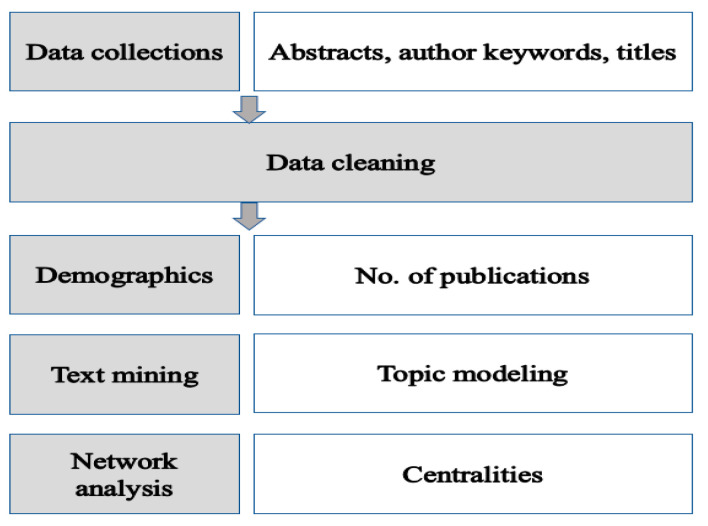
Study flowchart.

**Figure 2 healthcare-11-01139-f002:**
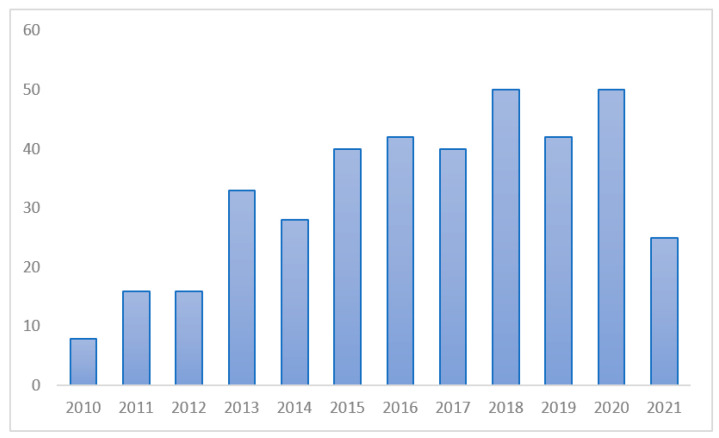
Number of publications.

**Figure 3 healthcare-11-01139-f003:**
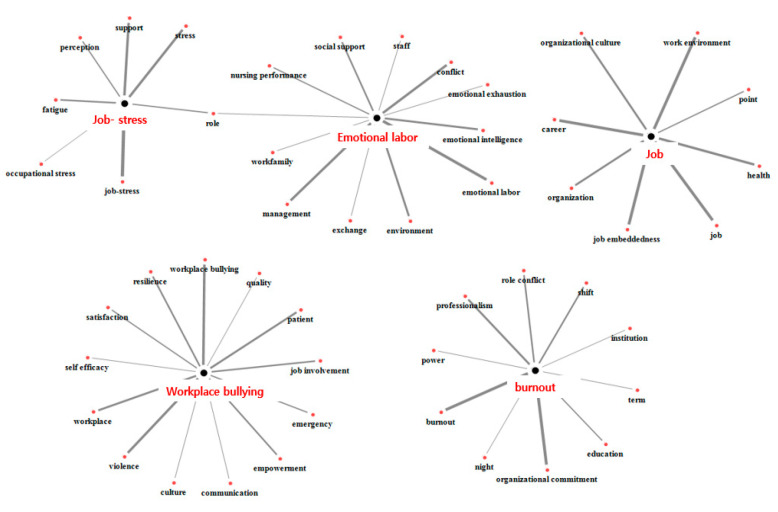
Visualization of the topic network.

**Figure 4 healthcare-11-01139-f004:**
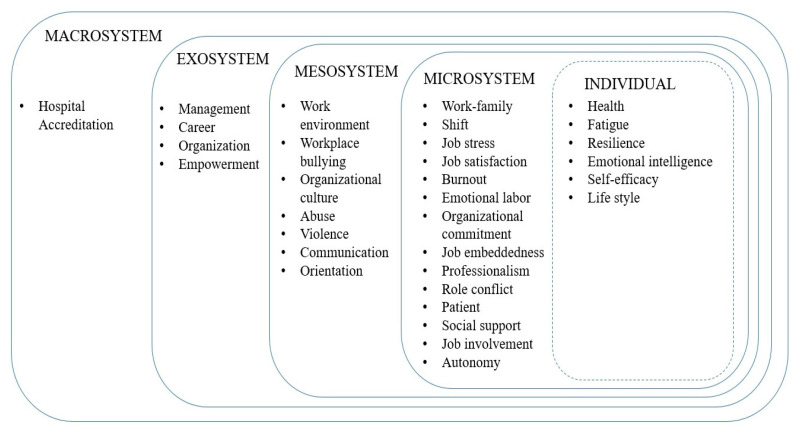
Classifications of the keywords by the Ecological Systems Theory.

**Table 1 healthcare-11-01139-t001:** Topic name and keywords.

Topic Name	No. of Articles N (%)	1st Keyword	2nd Keyword	3rd Keyword	4th Keyword	5th Keyword
Job	87 (22.3)	job	job embeddedness	work environment	career	health
Emotional labor	60 (15.4)	emotional labor	management	conflict	environment	emotional intelligence
Job stress	77 (19.7)	job stress	stress	support	fatigue	perception
Workplace bullying	82 (21.0)	workplace bullying	violence	workplace	patient	empowerment
Burnout	84 (21.5)	burnout	organizational commitment	professionalism	shift	role conflict
Total	390 (100)					

**Table 2 healthcare-11-01139-t002:** Frequencies, Degree Centrality, Betweenness Centrality, and Closedness Centrality of the Top 30 Keywords.

	Keywords	Frequencies	Degree Centrality	Betweenness Centrality	Closedness Centrality
1	**job stress**	311	*job satisfaction*	*job satisfaction*	**job stress**
2	**burnout**	232	**job stress**	**job stress**	**burnout**
3	**organizational commitment**	228	**burnout**	**emotional labor**	*job satisfaction*
4	**emotional labor**	215	**emotional labor**	*work*	**emotional labor**
5	**job**	202	*work*	**burnout**	**organizational commitment**
6	**job embeddedness**	145	**job**	**job**	*work*
7	work environment	138	**organizational commitment**	**organizational commitment**	**job**
8	career	134	**job embeddedness**	*experience*	*experience*
9	**stress**	114	*experience*	**conflict**	**job embeddedness**
10	**workplace bullying**	105	**stress**	**workplace bullying**	**stress**
11	management	102	empowerment	patient	empowerment
12	health	97	**professionalism**	culture	**fatigue**
13	violence	91	**conflict**	healthcare accreditation	**organizational culture**
14	**conflict**	90	**role conflict**	**stress**	environment
15	support	89	*satisfaction*	perception	*Work–family*
16	organization	81	social support	organization	work environment
17	**professionalism**	80	career	violence	**workplace bullying**
18	environment	77	emotional intelligence	health	abuse
19	**organizational culture**	76	**fatigue**	occupational stress	*satisfaction*
20	emotional intelligence	72	patient	communication	**role conflict**
21	workplace	71	nursing work environment	workplace	job involvement
22	role	70	**organizational culture**	**professionalism**	social support
23	patient	64	role	**job embeddedness**	self-efficacy
24	shift	59	work environment	lifestyle	**professionalism**
25	**role conflict**	58	*Work–family*	*satisfaction*	autonomy
26	empowerment	58	**workplace bullying**	**fatigue**	violence
27	**fatigue**	57	abuse	*Work–family*	**conflict**
28	social support	56	environment	**organizational culture**	degree
29	resilience	53	health	**role conflict**	resilience
30	job involvement	53	job involvement	orientation	nursing performance

Bold: Words that were included in the frequency analysis and all three types of centrality analyses—degree centrality, betweenness centrality, and closeness centrality. Italics: Words that were not included in the frequency analysis but were included in all three types of centrality analyses—degree centrality, betweenness centrality, and closeness centrality.

## Data Availability

Not applicable.

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
