# Peer review of "Research Topic Trends on Turnover Intention among Korean Registered Nurses: An Analysis Using Topic Modeling"

_healthcare, 2023, doi:10.3390/healthcare11081139_

Round 1

Reviewer 1 Report

Thanks for the opportunity to review this interesting paper, "Research Trends on Turnover Intention Among Korean Registered Nurses: An Analysis Using Topic Modeling," which addresses a relevant topic in health care globally.

The manuscript is theoretically grounded, logical, and parsimonious, highlighting its main aim and the lack of literature that justifies its relevance. The paper is well written, easy to read, and captures the reader's attention, showing an appropriate method design, which is clear and deeply described, in terms of data analysis, although regarding data collection proceedings, it needs clarification. It is not clear which databases the authors used/selected to extract the papers, and also, the use of the word "domestic" is less appropriate, so it must be changed or explained its meaning in the research. The results, the discussion, and the conclusion sections reveal a comprehensive style, linking the purpose of the study with theory and the main implications for practice. Findings are globally relevant for and reinforce theory, research, and practice - considering the world's demographic changes, primarily towards an aging population, nurse turnover is a preoccupancy topic. The limitations of the study are well stablished.

Author Response

The reviewer’s comments were very helpful overall, and we are appreciative of this constructive feedback on our submission.

Reviewer 2 Report

 Investing underlying issues contributing to turnover intention is a timely and relevant undertaking and is the focus of this manuscript.  That said, there are areas where clarification or further information would strengthen the manuscript.

A general comment:  The use of the term “research trends” can contribute to reader confusion, as one could easily think “research” refers to the design or methodology rather than what is being reported in this paper.  It seems authors explored what terms/concepts/variables have been investigated in relation to the turnover intention.  Using wors such as concepts or variables would help the reader understand the focus and findings better, particularly when the word “topic” is also used to describe the data.  So clarifying terms/words is critical to enhancing clarity.

Reason for undertaking the study:  Lines 68-71:  Please strengthen this section as it is not clear what additional factors should be examined given the plethora of literature on turnover intention.  Using words such as “crucial” may be a little too strong.  Clarifying the gap being addressed will strengthen the manuscript.

Line 85:  Please provide a reference when first discussing text network analysis.

Sources:  Please provide rationale for not including CINAHL and Medline as data bases to be searched, given the international focus.

Data analysis: Please provide rationale for limiting the number of topics – themes?- to five.  In addition, please provide discussion of what the numbers mean in relation to degree centrality, betweenness,  and closedness centrality.  Otherwise, the numbers are irrelevant.  Provide an example of one of the major classifications, in addition to the visualization of the topic network. 

Discussion, Line 184:  Please rephrase the first sentence, as the modeling did not identify the major topics, the investigators did. 

Please consider placing findings within a theoretical framework – organizational culture plays a big role in retention so think about how your findings fit with the larger context.

Lastly, make clear the contribution being provided by this study, as it would seem findings only support what is already known.  Perhaps discussing the universality of factors influencing turnover intention would help readers appreciate

Author Response

(The authors gave the same response as above.)

Reviewer 3 Report

This is an interesting work but I urge the authors to provide the following clarifications and arguments.

-        Incremental contribution and novelty of this study-why needed and how novel

-        Add a section on Korea and nursing issues for international audience

-        Provide clearer goals- just providing “data” is very vague and in need of specification

-        Page 2 lines 59-63 avoid starting two times consecutive sentences with “while”

-        Clearly define and argument inclusion and exclusion criteria (why 2010-2020 time frame and why all other decisions)

-        Define and explain for unfamiliar readers many terms, TF-IDF etc

-        What is the main message and finding from the study

-        What are the implications for Korean and other international contexts?

Author Response

(The authors gave the same response as above.)

Round 2

Reviewer 2 Report

Thanks for addressing reviewer concerns - much more clear now!